# An Appliance Scheduling System for Residential Energy Management

**DOI:** 10.3390/s21093287

**Published:** 2021-05-10

**Authors:** Hanife Apaydin-Özkan

**Affiliations:** Department of Electrical and Electronics Engineering, Eskisehir Technical University, Eskisehir 26555, Turkey; hapaydin1@eskisehir.edu.tr

**Keywords:** residential energy management, appliance scheduling, peak demand, user comfort, brute-force closest pair

## Abstract

In this work, an Appliance Scheduling-based Residential Energy Management System (AS-REMS) for reducing electricity cost and avoiding peak demand while keeping user comfort is presented. In AS-REMS, based on the effects of starting times of appliances on user comfort and the user attendance during their operations, appliances are divided into two classes in terms of controllability: MC-controllable (allowed to be scheduled by the Main Controller) and user-controllable (allowed to be scheduled only by a user). Use of all appliances are monitored in the considered home for a while for recording users’ appliance usage preferences and habits on each day of the week. Then, for each MC-controllable appliance, preferred starting times are determined and prioritized according to the recorded user preferences on similar days. When scheduling, assigned priorities of starting times of these appliances are considered for maintaining user comfort, while the tariff rate is considered for reducing electricity cost. Moreover, expected power consumptions of user-controllable appliances corresponding to the recorded user habits and power consumptions of MC-controllable appliances corresponding to the assigned starting times are considered for avoiding peak demand. The corresponding scheduling problem is solved by Brute-Force Closest Pair method. AS-REMS reduces the peak demand levels by 45% and the electricity costs by 39.6%, while provides the highest level of user comfort by 88%. Thus, users’ appliance usage preferences are sustained at a lower cost while their comfort is kept impressively.

## 1. Introduction

In the present day, the population and the usage of technological devices are increased in cities yielding an increase in energy demand. That high energy demand causes high depletion of natural resources and pollution of the environment as well as high costs for both users and energy providers. Hence, efficient and conscious use of energy is essential for people, the environment and the future. Since residential energy consumption constitutes 38% of the total energy consumption in the US [1], studies on Residential Energy Management (REM) have gained importance nowadays.

Residential users have various habits of energy use according to their lifestyles and want to keep their comfort in today’s life, while reducing electricity cost is the common goal of all users. Hence, keeping user comfort and reducing electricity cost are two parameters that should be under consideration in REM studies. On the other hand, the total electricity consumption of independent homes may exceed the power limit provided by the grid, thus the peak demand occurs at certain times of the day, i.e., in the evenings when all occupants are at home. This leads to expensive failures in the grid and the requirement for more grid infrastructure to prevent these failures. Grid malfunctions may also pose serious problems affecting the public’s social life, such as disruption of health and transportation activities in the city. Consequently, for both residential users and energy provider sides, avoiding peak demand parameters should also be considered in REM studies.

Many studies have been done on REM systems in the literature considering electricity cost, peak demand and user comfort.

Some REM studies in the literature dealt with cost reduction and keeping user comfort simultaneously. Through these studies, the work in [2] proposed a pre-emptive priority-based load scheduling approach at residential premises, while a REM algorithm using reinforcement learning and an artificial neural network was presented in [3]. The work in [4] demonstrated that comfort and energy consumption can be partially decoupled by an adaptive indoor comfort management approach. An automated switching off system with load balancing and appliance planning algorithm was proposed in [5]. In that work, all appliances are scheduled to manage the cumulative energy consumption below a defined power level with less interaction to users. Authors in [6] presented a multi-objective optimization model to reduce the electricity cost as well as the inconvenience level of the home user. They evaluated the performance of the proposed method by using the energy consumption patterns of several different social-economic Brazilian families. The work in [7] presented a consensual negotiation-based decision model for eliminating the overload by using appliances with the IoT concept. In that model, all connected appliances make their individual decisions based on the consensus algorithm. In [8], a REM approach was presented by a mixed-integer nonlinear programming problem with time or energy-based task classification. In [9] authors presented an improved multi-objective optimization algorithm to minimize the electricity cost with considering the user comfort. A new binary particle swarm optimization with quadratic transfer function was proposed in [10] for scheduling shiftable appliances in smarthomes. Authors in [11] presented a mathematical model to assist aggregator that is able to match a flexibility request from distributor system operator while reducing the cost and rescheduling shiftable appliances. In [12], a level billing approach was proposed with the aim of providing user comfort and cost reduction while a probabilistic scenario-based method [13] and an intensive quadratic programming approach [14] were presented with the same aim. Performance of different types of Demand Side Managements (DSMs) are compared in [15]; such as, deterministic and stochastic DSMs, and day-ahead and real-time DSMs. The authors in [16] propose robust energy management for grid-connected and islanding microgrids by considering stochasticity over the active power injections from photovoltaic units, wind turbine units, and conventional demands. Authors presented a multilayer control mechanism in [17] and they proposed to use Tabu search for scheduling HVAC (heating, ventilation and air conditioning) system.

Some studies in the literature aimed at reducing the electricity cost and avoiding the peak demand as well as keeping users comfortable. For example, the aim of the work [18] is to minimize the energy cost and dissatisfaction of the customer by using different electricity tariffs (time of use (TOU), inclining block rate (IBR) and real-time pricing (RTP)). The work in [19] proposes an automatic control approach that reduces the peak demand of buildings as compared to manual control. An incentive-based energy optimization method is proposed [20] for scheduling a number of residential electric appliances of a residential community. Authors propose a crow search optimization algorithm in [21] for appliance scheduling with RTP tariff rate. In [22], mixed-integer quadratic programming problem is proposed to find the optimal energy scheduling of controllable loads as well as charging/discharging strategies of the energy storage systems and plug electric vehicles by considering renewable energy resources (RESs). Authors take the forecast uncertainty caused by the RESs energy profiles into account, as well as the users’ energy demand.

The main drawback of these works is using the average powers of appliances instead of their real power profiles. That is, power consumption of appliances are assumed to be constant during a time period, i.e., 1 h. This drawback was eliminated in the authors’ previous works by using real power profiles of appliances: In [23], a real-time residential power management scheme based on power unit prioritization due to their current status and tariff rates was presented, while in [24] an appliance-based residential power management system that manages home’s power consumption based on the operational characteristics of smart appliances was introduced. Although user comfort was also taken into consideration besides reducing the electricity cost and avoiding the peak demand in these works, ignoring user preferences and allowing comprehensive intervention to some appliances kept the user comfort at a limited level.

Remarkable REM studies in the literature are summarized with their methods, objectives and descriptions in Table 1.

In this study, an Appliance Scheduling-based Residential Energy Management System (AS-REMS) which avoids peak demand and keeps user comfort while reducing electricity cost is proposed. In AS-REMS, based on the effects of starting times of appliances on user comfort and the user attendance during their operations, appliances are classified as MC-controllable appliances which are allowed to be scheduled by the Main Controller and user-controllable appliances which are allowed to be scheduled only by users. Use of all appliances are monitored in the considered home for a while for getting users’ appliance usage preferences and habits for each day of the week. Then for each MC-controllable appliance, favorite starting times are determined and prioritized according to the recorded user preferences on similar days. When scheduling MC-controllable appliances, assigned priorities of starting times are considered for maintaining user comfort. On the other hand, the sum of expected power consumption of user-controllable appliances corresponding to the recorded user habits, and the power consumption of MC-controllable appliances corresponding to the assigned starting times is obtained as the total power consumption of considered home which is taken into account for avoiding peak demand, while the tariff rate is considered for reducing the electricity cost. The corresponding scheduling problem is solved by Brute-Force Closest Pair method.

AS-REMS provides important advantages over similar REM studies. The main contributions of AS-REMS to the literature can be summarized as follows:AS-REMS is a multi-objective REMS structure that considers avoiding peak demand, reducing electricity cost and keeping user comfort simultaneously.AS-REMS provides a realistic and high-level user comfort; because it is based on the users’ appliance usage preferences and habits which are obtained by monitoring the considered home for a while.AS-REMS assures to detect the short-term peak demand and consequently procure smooth and continuous energy from the grid since it uses real power consumptions of appliances instead of their rated (average) powers.

This paper is organized as follows: The proposed AS-REMS is introduced in Section 2 in detail. Case studies and their results are presented and interpreted in Section 3. Finally, conclusions and future work directions are given in Section 4.

## 2. Appliance Scheduling System for Residential Energy Managements

In this work, AS-REMS is proposed for scheduling allowed appliances with the aim of avoiding peak demand and reducing electricity cost while keeping user comfort.

AS-REMS consists of the Main Controller (MC), a database, communication units, electrical appliances, power measurement units (smart plugs) mounted on appliances, control units (Wifi-RS232 converters or Wifi-relay modules) and a smart meter. The configuration of AS-REMS is presented in Figure 1.

In AS-REMS, one execution period (e.g., one day, 24 hours) is discretized into a prescribed *T* number of uniform time slots, i.e., t∈T={1,2,…,T}; hence, the total number of time slots (shortly, ts) in a day is $T=24\Delta_{t}$. Here, $\Delta_{t}$ represents the length of each ts.

### 2.1. Appliances

In AS-REMS, appliance scheduling is strictly based on users’ appliance usage preferences. Within this scope, the home is monitored for a while to constitute usage and power consumption information of appliances. During the monitoring, at the beginning of each ts of one execution period T (e.g., one day), MC communicates with appliances to gather usage information of each appliance a∈L, where L represents the set of appliances. This appliance usage information is stored in a database in a matrix form, namely utilization matrix. In AS-REMS, for each appliance a∈L distinct utilization matrices are composed for each set of similar days of observed weeks; thus seven different utilization matrices are constructed for each appliance. The set of similar days of the observed weeks is represented by the set *D*.

Utilization matrix of an appliance a∈L for the set of the similar days *D*, is represented by $U_{D}^{a}:{\left\{0,1\right\}}^{|D|\times T}$. $U_{D}^{a}\left(d,t\right)$ is constructed as in Equation (Equation 1):(1)$$U_{D}^{a}\left(d,t\right)=\left\{\begin{array}{ccc} 1 & , & a\;is\;operating\;at\;ts\;t\in T\;on\;d\in D\\ 0 & , & o.w. \end{array}\right.$$

In AS-REMS, in order to get power consumption profiles of appliances, power consumptions of appliances are measured via power measurement units during proper durations and stored in a database in a vector form, namely power profile vector. Power profile vector of an appliance a∈L is represented by ${\tilde{P}}^{a}:R^{1\times |T_{m}^{a}|}$, where $T_{m}^{a}\in T$ is the measurement duration of *a* and ${\tilde{P}}^{a}\left(\tilde{t}\right)$ refers to the power consumption of *a* at its $\tilde{t}$th internal ts. Note that, in this study, the internal ts of appliances during their operation is indicated by $\tilde{t}$ ($\Delta_{\tilde{t}}=\Delta_{t}$); such that $\tilde{t}=0$ at the time that the appliance is turned on, $\tilde{t}$ increases as long as the appliance is running, $\tilde{t}$ is reset when the appliance is switched off.

In AS-REMS, power consumption of an appliance a∈L at a ts t∈T of a day d∈D is defined in Equation (Equation 2).
(2)$$P_{d}^{a}\left(t\right)={\tilde{P}}^{a}\left(t-t_{s}^{a}\right)U_{D}^{a}\left(d,t\right)$$

Here, $t_{s}^{a}\in T$ is the starting time of the appliance *a*, and $t-t_{s}^{a}$ refers to internal ts $\tilde{t}$ of *a*.

In appliance scheduling-based REM studies, generally, it is needed to interfere with appliances externally. It is not appropriate for some appliances because of their own intended use and technical features. Therefore, most of the REM studies dealing with appliance scheduling in the literature have considered the classification of appliances. The main basis of classifications is the suitability of appliances for external interference. Therefore, classification types are consistent with each other, although assigned class names are different; such as controllable (C)/uncontrollable (UnC), shiftable (Sh)/unshiftable (USh), schedulable (Sc)/unschedulable (USc), normally operated (NO), fixed and task-based (FTB), comfort-based elastic (CBE), energy-based elastic (EBE) etc. (see Table 2 for types of classification in the literature). For example, the refrigerator is considered in categories such as unshiftable, uncontrollable or task-based, since its operation time and duration are not suitable for any external interference.

In AS-REMS, the suitability of appliances for external interference is determined based on the effects of starting times of appliances on user comfort and user attendance during their operations. Accordingly, appliances are divided into two classes in terms of controllability: MC-controllable (MCC) appliances which are allowed to be scheduled by a main controller and user-controllable (USC) appliances which are allowed to be scheduled only by a user. The set of appliances is represented by $L=L_{\mathrm{MC}}\cup L_{\mathrm{UC}}$, where $L_{\mathrm{MC}}$ is the set of MC-controllable appliances, while $L_{\mathrm{UC}}$ is the set of user-controllable appliances. These classes will be explained in detail in the following subsections.

#### 2.1.1. User-Controllable Appliances

Appliances whose starting times directly affect user comfort are classified as user-controllable appliances. Their starting times are set by users and are not negotiable. Interfering with starting times of these appliances against the demand of users undoubtedly deteriorates user comfort. User-controllable appliances can be two types: non-delayable (ndUSC) and delayable (dUSC). Non-delayable user-controllable appliances are generally appliances that must be turned on immediately upon users’ request, and they are basically operated by an attending user (e.g., TV, hairdryer, toaster, rice cooker, microwave oven, vacuum cleaner, iron, lights and etc.). Appliances whose operations are fixed (e.g., refrigerator) are also considered in this type. On the other hand, delayable user-controllable appliances can be scheduled for a specific time due to users’ requests (e.g., kettle, coffee machine, water heater, air-conditioner). For example, when a user wants coffee to be ready at 8:00 a.m., he/she can schedule the starting time of the coffee machine correspondingly. For both delayable and non-delayable user-controllable appliances, only users can decide when and how long these appliances will operate. Hence, any user-controllable appliances are not allowed to be scheduled by MC in AS-REMS.

The power consumption of a user-controllable appliance $a\in L_{\mathrm{UC}}$ is given in Equation (Equation 3).
(3)$$P_{\mathrm{UC}}^{a}\left(t\right)={\tilde{P}}^{a}\left(t-t_{s}^{a}\right)U_{D}^{a}\left(d,t\right)$$

Here,  $\forall t\in [t_{s}^{a}-T_{m}^{a}]$. In AS-REMS, power measurement duration $T_{m}^{a}$ of any user-controllable appliance $a\in L_{\mathrm{UC}}$ is one execution period, that is $T_{m}^{a}=T$. Power consumption profiles of a kettle and an air-conditioner are given as examples of power consumption profiles of user-controllable appliances in Figure 2 and in Figure 3, respectively.

#### 2.1.2. MC-Controllable Appliances

Appliances whose starting times can interfere without deteriorating user comfort are classified as MC-controllable appliances. These appliances are unattended appliances that are operated with little supervision (e.g., washing machine, dishwasher, tumble dryer, battery-powered appliances). For example, dirty laundry can wait in the washing machine for a while (until the assigned starting time) without deteriorating user comfort. Hence, MC-controllable appliances are allowed to be scheduled by MC in AS-REMS.

Any MC-controllable appliance $a\in L_{MC}$ operates during a certain time $T_{o}^{a}\in T$ after it is switched on. The power consumption of a MC-controllable appliance $a\in L_{MC}$ is given in Equation (Equation 4).
(4)$$P_{MC}^{a}\left(t\right)={\tilde{P}}^{a}\left(t-t_{s}^{a}\right)U_{D}^{a}\left(d,t\right)\;\;\forall t\in \left[t_{s}^{a}-T_{m}^{a}\right]$$

In AS-REMS, power measurement duration $T_{m}^{a}$ of any MC-controllable appliance $a\in L_{MC}$ is its operation period, that is $T_{m}^{a}=T_{o}^{a}$. The power consumption profiles of a washing machine is given in Figure 4.

### 2.2. Brute Force Closest Pair Method

Brute-Force Closest Pair (BFCP) method finds the closest point to a reference point through a set of candidate points by considering euclidean distance. For example, let we consider Figure 5 where $\left(x_{r},y_{r}\right)$ is the reference point and the other $\left(x_{i},y_{i}\right),\;i\in \left\{1,2,\ldots ,5\right\}$ are candidate points making up the set $P_{candidates}$.

BFCP finds out all euclidean distances of the reference point $P_{r}=\left(x_{r},y_{r}\right)$ from the five points ${P_{c}}_{i}=\left(x_{i},y_{i}\right),\;i\in \left\{1,2,..,5\right\}$ which accumulates 5 distance computations {$P_{r}P_{c_{1}}$, $P_{r}P_{c_{2}}$, $P_{r}P_{c_{3}}$, $P_{r}P_{c_{4}}$, $P_{r}P_{c_{5}}$}, and determines the green point as the closest point from the reference point according to the following euclidean distance equation:(5)$$Distance=\min_{\left(x_{i},y_{i}\right)\in P_{candidates}}\sqrt{\left({\left(x_{i}-x_{r}\right)}^{2}+{\left(y_{i}-y_{r}\right)}^{2}\right)}$$

### 2.3. Scheduling Parameters

AS-REMS schedules MC-controllable appliances at the beginning of each day with the aims of avoiding peak demand and reducing electricity cost while keeping user comfort. Hence the scheduling parameters are electricity cost, peak demand and user comfort.

#### 2.3.1. User Comfort

For scheduling MC-controllable appliances without deteriorating user comfort, AS-REMS considers users’ appliance usage preferences stored in the database. For each MC-controllable appliance $a\in L_{MC}$, starting times $t_{s}^{a}$ on similar days in *D* are obtained from the corresponding utilization matrix $U_{D}^{a}$ and listed in the set of starting times, i.e., $TS_{D}^{a}$ as in Equation (Equation 6).
(6)$$TS_{D}^{a}=\left\{t_{s}^{a}|U_{D}^{a}\left(d,t_{s}^{a}\right)-U_{D}^{a}\left(d,\left(t_{s}^{a}-1\right)\right)=1\;,\;t_{s}^{a}\in T\;,\;d\in D\right\}$$

For each $t_{s}^{a}\in TS_{D}^{a}$, the number of being chosen as starting time, i.e., $NC_{D}^{a}\left(t_{s}^{a}\right)$, and its probability, i.e., $Pr_{D}^{a}\left(t_{s}^{a}\right)$, at any day in *D* are calculated in Equation (Equation 7) and Equation (Equation 8) respectively.
(7)$$NC_{D}^{a}\left(t_{s}^{a}\right)=\sum_{d\in D}\left(U_{D}^{a}\left(d,t_{s}^{a}\right)-U_{D}^{a}\left(d,\left(t_{s}^{a}-1\right)\right)\right)$$
(8)$$Pr_{D}^{a}\left(t_{s}^{a}\right)=\frac{NC_{D}^{a}\left(t_{s}^{a}\right)}{\sum_{t_{s}^{a}\in TS_{D}^{a}}NC_{D}^{a}\left(t_{s}^{a}\right)}$$

Then for each appliance $a\in L_{\mathrm{MC}}$ each starting time $t_{s}^{a}$ is labeled with the corresponding priority level for the considered day *d*, i.e., ${Pr_{L}}_{d}^{a}\left(t_{s}^{a}\right)$, such that the priority level of starting time with the highest probability is 1, that with the second-highest probability is 2, and so on. Note that, the priority level of the starting time with the lowest probability is $|TS_{D}^{a}|$.

The total priority level induced by starting the operation of MC-controllable appliances $a_{i}\in L_{MC}$ at times $t_{s}^{a_{i}}\in TS_{D}^{a_{i}}$ in a day d∈D is defined as the square root of the sum of the squared priority level of each MC-controllable appliance as in Equation (Equation 9).
(9)$${Pr_{L}}_{d}^{L_{MC}}\left(T_{s}^{L_{MC}}\right)=\sqrt{\sum_{a_{i}\in L_{MC}}{\left({Pr_{L}}_{d}^{a_{i}}\left(t_{s}^{a_{i}}\right)\right)}^{2}}$$

Here, $T_{s}^{L_{MC}}$ stands for a combination of starting times of all MC-controllable appliances, such that $\left(t_{s}^{a_{1}},t_{s}^{a_{2}},\ldots ,t_{s}^{a_{|L_{MC}|}}\right)$ where $t_{s}^{a_{i}}\in TS_{D}^{a_{i}},\;\forall a_{i}\in L_{MC}$.

Unlike previous studies in the literature, for a more realistic approach, AS-REMS gets users’ appliance usage preferences by monitoring their power consumption in the considered home. The total priority level is a determining parameter that shows the preference of starting time combinations of MC-controllable appliances. As operating an appliance at the most preferred starting time increases user comfort, AS-REMS considers operating the MC-controllable appliances at the most preferred starting times by minimizing the total priority level.

Users can prefer to operate MC-controllable appliances at times different from the recorded user habits, which may yield uncertainties at the preferred starting times. In order to eliminate the effects of these uncertainties on user comfort, users are also allowed to select a specific starting time interval $T_{s_{interval}}^{a_{i}}\subset T$ for each MC-controllable appliance $a_{i}\in L_{MC}$. In this case, the user’s present preference is considered instead of stored historical usage preferences, and the priority level of $a_{i}$ is set to 0 (i.e., ${Pr_{L}}_{d}^{a_{i}}\left(t\right)=0$ ∀t∈T). Therefore, the priority level of $a_{i}$ does not add up to the total priority level value.

#### 2.3.2. Electricity Cost

Since the electricity tariff rate is generally time-dependent, different starting times of appliances yield different electricity costs. For scheduling MC-controllable appliances with reducing the electricity cost, AS-REMS takes the tariff rate into consideration.

The total electricity cost of MC-controllable appliances $a_{i}\in L_{MC}$ induced by starting their operation at times $t_{s}^{a_{i}}\in TS_{D}^{a_{i}}$ in a day d∈D is calculated in Equation (Equation 10).
(10)$$C_{d}^{L_{MC}}\left(T_{s}^{L_{MC}}\right)=\sum_{a_{i}\in L_{MC}}\sum_{t\in T}{\tilde{P}}^{a}\left(t-t_{s}^{a_{i}}\right)U_{D}^{a_{i}}\left(d,t\right)\mathrm{Tariff}\left(t\right)$$

Here, Tariff (t) is the unit price of electricity per kWh at a ts *t*.

For reducing electricity cost, AS-REMS considers minimizing the total electricity cost as much as possible.

#### 2.3.3. Peak Demand

For scheduling MC-controllable appliances by avoiding peak demand, AS-REMS intends total power consumption of appliances in the considered home not to exceed previously specified grid power limit, $P_{lim}$, at any time of the day. The total power consumption at each time is the sum of the expected power consumption of user-controllable appliances corresponding to the recorded user habits on similar days, and the power consumption of MC-controllable appliances corresponding to the assigned starting times.

Expected power consumption of a user-controllable appliance $a\in L_{UC}$ at a ts *t* of a day d∈D, i.e., $P_{d_{exp}}^{a}\left(t\right)$, is calculated in Equation (Equation 11) by regarding all similar days, that is, all days in *D*:(11)$$P_{d_{exp}}^{a}\left(t\right)=\frac{\sum_{d\in D}P_{d}^{a}\left(t\right)}{\left|D\right|}.$$

Expected power consumption of all user-controllable appliances at a ts *t* of a day d∈D, i.e., $P_{d_{exp}}^{L_{UC}}\left(t\right)$, is calculated in Equation (Equation 12).
(12)$$P_{d_{exp}}^{L_{UC}}\left(t\right)=\sum_{a\in L_{UC}}P_{d_{exp}}^{a}\left(t\right).$$

Power consumption of MC-controllable appliances $a_{i}\in L_{MC}$ with starting times $t_{s}^{a_{i}}\in TS_{D}^{a_{i}}$ in a day d∈D at a ts t∈T, i.e., $P_{d}^{L_{MC}}\left(t,T_{s}^{L_{MC}}\right)$, is calculated in Equation (Equation 13).
(13)$$P_{d}^{L_{MC}}\left(t,T_{s}^{L_{MC}}\right)=\sum_{a_{i}\in L_{MC}}{\tilde{P}}^{a_{i}}\left(t-t_{s}^{a_{i}}\right)U_{D}^{a_{i}}\left(d,t\right)$$

Corresponding total power consumption of all appliances in a day d∈D at a ts t∈T is calculated as in Equation (Equation 14).
(14)$$P_{d}^{L}\left(t,T_{s}^{L_{MC}}\right)=P_{d_{exp}}^{L_{UC}}\left(t\right)+P_{d}^{L_{MC}}\left(t,T_{s}^{L_{MC}}\right).$$

According to the total power consumption, whether the predefined power limit, $P_{lim}$, is exceeded at any ts t∈T of the day d∈D is represented by power limit indicator, i.e., $I_{d}\left(T_{s}^{L_{MC}}\right)$ is obtained as in Equation (Equation 15).
(15)$$I_{d}\left(T_{s}^{L_{MC}}\right)=\left\{\begin{array}{ccc} 0 & , & P_{lim}-P_{d}^{L}\left(t,T_{s}^{L_{MC}}\right)>0\;\forall t\in T\\ 1 & , & o.w \end{array}\right.$$

In order to provide a realistic approach, AS-REMS uses recorded appliance usage habits to determine expected power consumptions of user-controllable appliances and consequently Power limit indicator which is a determining parameter that shows power limit violation of power consumption. For avoiding the peak demand, AS-REMS deals to keep power limit indicator value at 0.

Users can operate user-controllable appliances whenever they want which may yield uncertainities at the total power consumption. In order to eliminate the effects of these uncertainities at the electricity cost and power limit indicators, these parameters are calculated by considering the expected power consumptions which are determined by usage of appliances during several days under several environmental conditions.

### 2.4. Scheduling Procedure

AS-REMS aims to schedule MC-controllable appliances by minimizing power limit indicator, total electricity cost and total priority level parameters. The corresponding scheduling procedure is given in AS-REMS Algorithm (namely, Algorithm 1).

At step 1 of AS-REMS Algorithm, the set of starting times $TS_{D}^{a_{i}}$ is found for each appliance $a_{i}\in L_{MC}$. If user does not select a specific starting time interval $T_{s_{interval}}^{a_{i}}=\emptyset$ for an appliance $a_{i}\in L_{MC}$, $TS_{D}^{a_{i}}$ is obtained as given in Equation (Equation 6), otherwise $TS_{D}^{a_{i}}$ is the set of all times in $T_{s_{interval}}^{a_{i}}$ and the priority level of $a_{j}$ does not add up to the total priority level value (i.e., ${Pr_{L}}_{d}^{a_{i}}\left(t\right)=0$ ∀t∈T).

At step 2 of AS-REMS Algorithm, the set of all possible combinations of starting times of MC-controllable appliances, namely practical solution set, is constructed for the considered day d∈D as follows:(16)$$TS_{D}^{L_{MC}}=\left\{T_{s_{1}}^{L_{MC}},\;T_{s_{2}}^{L_{MC}},\ldots ,\left(T_{s_{n}}^{L_{MC}}\right)\right\}$$
where $T_{s_{j}}^{L_{MC}}=\left\{t_{s}^{a_{1}},\;t_{s}^{a_{2}},\ldots ,\;t_{s_{j}}^{a|L_{MC}|}\right\}$ is the $j^{th}$ possible starting time combination and *n* is the number of the possible combinations $n=\underset{\forall a_{i}\in L_{MC}}{\prod }\left|TS_{D}^{a_{i}}\right|$.

At step 3 of AS-REMS Algorithm, for each practical solution $T_{s_{j}}^{L_{MC}}\in TS_{D}^{L_{MC}}$ parameters of total electricity cost $C_{d}^{L_{MC}}\left(T_{s_{j}}^{L_{MC}}\right)$, power limit indicator $I_{d}\left(T_{s_{j}}^{L_{MC}}\right)$ and total priority level ${Pr_{L}}_{d}^{L_{MC}}\left(T_{s_{j}}^{L_{MC}}\right)$ are calculated.

In order to find the optimal practical solution among the candidate solution set, the BFCP method which uses the Euclidean distance approach to find the optimal solution through a practical solution set is used. Note that, the number of MC-controllable appliances and the number of their preferred starting times are limited, the number of possible combinations of starting times of MC-controllable appliances, thus the size of the practical solution set is also limited in this problem. Hence, applying the BFCP method (calculating the value of the objective function for each possible practical solution and which is also verified by the analysis results given in the case study section.

**Algorithm 1** AS-REMS algorithm.
**Input:**$L_{MC}$, $L_{UC}$, *d*, $T_{s_{interval}}^{a_{i}}$,  $U_{D}^{a}$, ${\tilde{P}}^{a}$, $P_{lim}$, Tariff.**Step 1.** Determine the set of possible starting times $TS_{D}^{a_{i}}$ of each $a_{i}\in L_{MC}$ such that:      If $T_{s_{interval}}^{a_{i}}=empty$            Find $TS_{D}^{a_{i}}$ via Equation (Equation 6)      Else            $TS_{D}^{a_{i}}=T_{s_{interval}}^{a_{i}}$            ${Pr_{L}}_{d}^{a_{i}}\left(t_{s}^{a_{i}}\right)=0$      End If**Step 2.** Determine the set of possible combination of possible starting times MC-controllable appliances $TS_{D}^{L_{MC}}$ as in Equation (Equation 16)**Step 3.** Determine corresponding parameters values for each $T_{s_{j}}^{L_{MC}}\in TS_{D}^{L_{MC}}$           ${Pr_{L}}_{d}^{L_{MC}}\left(T_{s_{j}}^{L_{MC}}\right)$ via Equation (Equation 9)           $C_{d}^{L_{MC}}\left(T_{s_{j}}^{L_{MC}}\right)$ via Equation (Equation 10)           $I_{d}\left(T_{s_{j}}^{L_{MC}}\right)$ via Equation (Equation 15)**Step 4.** Normalize the corresponding parameter values for each $T_{s_{j}}^{L_{MC}}\in TS_{D}^{L_{MC}}$ as          ${Pr_{L}}_{d}^{L_{MC}}\left(T_{s_{j}}^{L_{MC}}\right)\to {Pr_{L}}_{d}^{L_{MC}}\left(T_{s_{i}}^{L_{MC}}\right)$          $C_{d}^{L_{MC}}\left(T_{s_{j}}^{L_{MC}}\right)\to C_{dscaled}^{L_{MC}}\left(T_{s_{i}}^{L_{MC}}\right)$          $I_{d}\left(T_{s_{j}}^{L_{MC}}\right)\to I_{dscaled}\left(T_{s_{j}}^{L_{MC}}\right)$**Step 5.** Call BFCP Algorithm with ${Pr_{L}}_{d}^{L_{MC}}\left(T_{s}^{L_{MC}}\right)$, $I_{dscaled}\left(T_{s}^{L_{MC}}\right)$, $C_{dscaled}^{L_{MC}}\left(T_{s}^{L_{MC}}\right)$
**Output:**
$T_{s_{optimal}}^{L_{MC}}$



Since BFCP method uses euclidean distance approach and takes the magnitude of parameters neglecting the units, parameters with high magnitude ranges will dominate the parameters with low magnitude ranges. In order to supress this effect and each of parameter to contribute to the result equally, all parameters are brought to the same scale of magnitudes at step 4 of AS-REMS Algorithm. Thus, total electricity cost of MC-controllable appliances $C_{d}^{L_{MC}}\left(T_{s}^{L_{MC}}\right)$, and power limit indicator values $I_{d}\left(T_{s}^{L_{MC}}\right)$ are scaled in the range of priority level values ${Pr_{L}}_{d}^{L_{MC}}\left(T_{s}^{L_{MC}}\right)$ via min-max normalization yielding scaled total electricity cost values, i.e., $C_{dscaled}^{L_{MC}}\left(T_{s}^{L_{MC}}\right)$, and scaled power limit indicator values, i.e., $I_{dscaled}\left(T_{s}^{L_{MC}}\right)$, respectively.

At step 5 of AS-REMS Algorithm, BFCP with the normalized parameters of ${Pr_{L}}_{d}^{L_{MC}}$$\left(T_{s}^{L_{MC}}\right)$, $I_{dscaled}\left(T_{s}^{L_{MC}}\right)$ and $C_{dscaled}^{L_{MC}}(T_{s}^{L_{MC}}$ is applied. BFCP searches the optimal starting time combination of MC-controllable appliances, i.e., $T_{s_{opt}}^{L_{MC}}\in TS_{D}^{L_{MC}}$ such that ($C_{dscaled}^{L_{MC}}$$\left(T_{s_{opt}}^{L_{MC}}\right)$, $I_{dscaled}\left(T_{s_{opt}}^{L_{MC}}\right)$, ${Pr_{L}}_{d}^{L_{MC}}\left(T_{s_{opt}}^{L_{MC}}\right)$) which is the closest triple to the triple of ideal minimum parameter values according to the following euclidean objective function (Equation 17): (17)$$\begin{array}{cc} d_{i}= & sqrt(w_{Pr}.{\left({Pr_{L}}_{d}^{L_{MC}}(T_{s_{i}}^{L_{MC}}\right)-{{\hat{Pr}}_{L}}_{d}^{L_{MC}})}^{2}\\ \;\;\;\; & \;\;+w_{C}.{\left(C_{dscaled}^{L_{MC}}\left(T_{s_{i}}^{L_{MC}}\right)-{\hat{C}}_{d}^{L_{MC}}\right)}^{2}+w_{I}.{\left(I_{dscaled}\left(T_{s_{i}}^{L_{MC}}\right)-{\hat{I}}_{d}\right)}^{2}) \end{array}$$
where, $T_{s_{i}}^{L_{M}}$ represents *i*-th possible combination of starting times, ${\hat{Pr_{L}}}_{d}^{L_{MC}}$ stands for the ideal value of total priority level, ${\hat{C}}_{d}^{L_{MC}}$ stands for the ideal value of total electricity cost, ${\hat{I}}_{d}$ stands for the ideal value of power limit indicator. Note that for BFCP method of AS-REMS, ${\hat{Pr_{L}}}_{d}^{L_{MC}}=\sqrt{\left(\right|L_{MC}\left|\right)}$, ${\hat{C}}_{d}^{L_{MC}}$ = 0, ${\hat{I}}_{d}$ = 0 for $|L_{MC}|$ number of MC-controllable appliances.

Note that, by changing the weights of the parameters ($w_{C}$, $w_{I}$ and $w_{Pr}$) in the euclidean objective function (Equation 17), it is possible to find the optimal solution and the corresponding starting times of MC-controllable appliances for different weighted parameters. Thus, in the case that any parameter is desired to be more effective, this is achieved by increasing the weight of the corresponding parameter. For example, if the primary goal is reducing the electricity cost, the weight of the relevant parameter (i.e., $w_{C}$) is chosen bigger than the other weights (i.e., $w_{C}>w_{I}$, $w_{C}>w_{Pr}$). If the primary preference is keeping the user comfortable and reducing the electricity cost simultaneously, the weights of these two parameters (i.e., $w_{C}$ and $w_{Pr}$) are chosen higher than the weight of power limit indicator (i.e., $w_{C}>w_{I}$, $w_{Pr}>w_{I}$). If the weights of all parameters are selected equal (i.e., $w_{C}$ = $w_{I}$ = $w_{Pr}$ as in the scenarios of the case study), optimal starting times of MC-controllable appliances for equal precedence of three parameters are obtained.

Since the practical solution set $TS_{D}^{L_{MC}}$ of BFCP problem consists of only allowed solutions, any $T_{s_{opt}}^{L_{MC}}\in TS_{D}^{L_{MC}}$ minimizing the objective function is determined as the solution of the problem. Consequently, this problem is independent of any constraints.

## 3. Case Studies and Discussion

In this section, in order to demonstrate AS-REMS’s performance on avoiding peak demand and reducing electricity cost while keeping user comfort, several scenarios are designed and simulations of these scenarios are carried out.

The scenarios are for a residence monitored for 12 weeks. The residence is 120 m${}^{2}$ flat with four occupants and equipped with kettle (1000 W), hair dryer (1600 W), toaster (700 W), rice cookers (400 W), microwave ovens (800 W), vacuum cleaner (700 W), water heater (1000 W), iron (1700 W), coffee machine (500 W), TV (116 cm), lamps (25 W), refrigerator (nofrost-540 lt) and air conditioner (6.74 kW cooling and 7.03 kW heating capacity) as user controllable appliances and washing machine (wm) (7 kg front-load), dishwasher (dw) (60 cm free standing) and battery powered appliances (bp) (for example, e-scooter, e-bike, etc.) as MC-controllable appliances. The power consumptions of all appliances are measured via Itech IT9121 power meter and Fibaro smart wall plugs and measured real power consumption profiles of appliances are used in the experiments.

Some appliances (e.g., microwave ovens, dw, wm) can draw very high power in a very short time (<3 min). In order to catch these short term high power variations, time slot duration is taken as 2 min, i.e., $\Delta_{t}=2$ and T=720. Besides, the same days of the weeks are defined as similar days, that is for each appliance a∈L, seven different utilization matrices $U_{D}^{a}:{\left\{0,1\right\}}^{|12|\times 720}$ are constructed.

For the grid power, the time of use (TOU) pricing tariff rate set by the Turkish Electricity Distributor Company (TEDAS) is used [25]. This pricing tariff is a three-level TOU tariff with on-peak, mid-peak, and off-peak periods. As it is clear in Table 3, electricity prices are lower when the demand is low (off-peak) and higher when the demand is high (on-peak) to encourage the user. Besides, the grid power limit is chosen as $P_{lim}=4500$ W according to the agreement between home residents and TEDAS.

Let we consider scheduling wm, dw and bp on a Friday from June as Scenario 1. Thus, the set of similar days *D* is the set of monitored Friday days. The sets of starting times of wm, i.e., $TS_{D}^{wm}$, that of dw, i.e., $TS_{D}^{dw}$ and that of bp, i.e., $TS_{D}^{pb}$ are formed via utilization matrices, as follows:$$\begin{array}{c} TS_{D}^{\mathrm{wm}}=\left\{00:00,05:00,05:12,05.28,06:00,20:28,23:30,23:38\right\}\\ TS_{D}^{\mathrm{dw}}=\left\{16:30,16:40,17:04,19:12,20:00,22:00,22:32\right\}\\ TS_{D}^{\mathrm{bp}}=\left\{6:00,6:20,6:30,6:50,20:38,21:52\right\} \end{array}$$

For each possible preferred starting times of wm, i.e., $t_{s}^{\mathrm{wm}}\in TS_{D}^{\mathrm{wm}}$, that of dw, i.e., $t_{s}^{\mathrm{dw}}\in TS_{D}^{\mathrm{dw}}$ and that of bp, i.e., $t_{s}^{\mathrm{bp}}\in TS_{D}^{\mathrm{bp}}$, on similar days in *D*; the number of being chosen as starting time (i.e., $NC_{D}^{\mathrm{wm}}\left(t_{s}^{\mathrm{wm}}\right)$, $NC_{D}^{\mathrm{dw}}\left(t_{s}^{\mathrm{dw}}\right)$,$NC_{D}^{\mathrm{bp}}\left(t_{s}^{\mathrm{bp}}\right)$ and their probabilities (i.e., $Pr_{D}^{\mathrm{wm}}\left(t_{s}^{\mathrm{wm}}\right)$, $Pr_{D}^{\mathrm{dw}}\left(t_{s}^{\mathrm{dw}}\right)$, $Pr_{D}^{\mathrm{bp}}\left(t_{s}^{\mathrm{bp}}\right)$) and the corresponding priority levels of the considered day *d* (i.e., ${Pr_{L}}_{d}^{wm}\left(t_{s}^{\mathrm{wm}}\right)$, ${Pr_{L}}_{d}^{dw}\left(t_{s}^{\mathrm{dw}}\right)$) and ${Pr_{L}}_{d}^{bp}\left(t_{s}^{\mathrm{bp}}\right)$) are determined as given in Table 4, Table 5 and Table 6, respectively.

Moreover, for the considered day *d*, for each possible combinations of preferred starting times of these appliances, that is, for each possible triplet of ($t_{s}^{\mathrm{wm}},t_{s}^{\mathrm{dw}},t_{s}^{\mathrm{bp}}$) corresponding total electricity cost $C_{d}^{L_{MC}}\left(t_{s}^{wm},t_{s}^{dw},t_{s}^{bp}\right)$, power limit indicator values $I_{d}\left(t_{s}^{wm},t_{s}^{dw},t_{s}^{bp}\right)$ and total priority level values ${Pr_{L}}_{d}^{L_{MC}}\left(t_{s}^{\mathrm{wm}},t_{s}^{\mathrm{dw}},t_{s}^{\mathrm{bp}}\right)$ are determined and represented on the three-axis chart in Figure 6. For some numerical samples, see Table 7.

**Algorithm 2** BFCP Algorithm of AS-REMS**    Input:**${Pr_{L}}_{d}^{L_{MC}}\left(T_{s}^{L_{MC}}\right)$, $I_{dscaled}\left(T_{s}^{L_{MC}}\right)$, $C_{dscaled}^{L_{MC}}\left(T_{s}^{L_{MC}}\right)$    ${\hat{Pr_{L}}}_{d}^{L_{MC}}$, ${\hat{C}}_{d}^{L_{MC}}$, ${\hat{I}}_{d}$ = $\left(sqrt(\right|L_{MC}\left|),0,0\right)$    $d_{min}=\infty$    for i=1 to *N*         $d_{i}=sqrt(w_{Pr}.\left({Pr_{L}}_{d}^{L_{MC}}\left(T_{s_{i}}^{L_{MC}}\right)-{\hat{Pr_{L}}}_{d}^{L_{MC}}\right){)}^{2}+$                                                     ${\left(w_{C}.\left(C_{dscaled}^{L_{MC}}\left(T_{s_{i}}^{L_{MC}}\right)-{\hat{C}}_{d}^{L_{MC}}\right)\right)}^{2}+{\left(w_{I}.\left(I_{dscaled}\left(T_{s_{i}}^{L_{MC}}\right)-{\hat{I}}_{d}\right)\right)}^{2})$            If $d_{i}<d_{min}$                  $d_{min}=d_{i}$; index=i            End If    End For    **Output:**
$T_{s_{index}}^{L_{MC}}$; $d_{min}$


In the considered day, through the possible triplets of starting times, triplets with the cheapest cost (9.45 cent), their priority level values and power limit indicator values are given in Table 8 and indicated by blue diamond on the chart in Figure 6. For some of these triplets, the power limit is exceeded. For one of these triplets corresponding daily total power consumption graph is given Figure 7.

On the other hand, through the possible triplets of starting times, the triplet with the minimum total priority level value (i.e., 1.73) is (05:12, 16:40, 06:50) (indicated by a red square on the chart in Figure 6). The induced cost of this triplet is 13.44 cent, while the power limit is exceeded. The daily power consumption graph of this triplet is given in Figure 8.

In order to obtain optimal starting times of wm, dw and bp from the view of cost, peak demand and user comfort, for each possible triplet of starting times, AS-REMS scales corresponding total electricity cost values and power limit indicator values in the range of priority level values via min-max normalization yielding scaled total electricity cost, i.e., $C_{dscaled}^{L_{MC}}\left(t_{s}^{wm},t_{s}^{dw},t_{s}^{bp}\right)$, and scaled power limit indicator values, i.e., $I_{dscaled}\left(t_{s}^{wm},t_{s}^{dw},t_{s}^{bp}\right)$, respectively (see Table 7 for examples) and schedules starting time of wm at (05:12), i.e., $t_{s}^{wm}=$05:12, that of dw at (22:32) i.e., $t_{s}^{dw}=$22:32, and that of bp at (21:52), i.e., $t_{s}^{bp}=$ 21:52, according to BFCP Algorithm. For the corresponding triplet of starting times, i.e., (05:12,22:32,21:52) (marked as black circle on the chart in Figure 6), the cost is 9.45 cent, the total priority level value is 7.28, while the power limit is not exceeded, i.e., $I_{d}\left(t_{s}^{wm},t_{s}^{dw},t_{s}^{bp}\right)=0$. The daily power consumption graph is given in Figure 9. As it is clear from the Figure 6, this is the closest triplet to the theoretically ideal point.

Let we consider scheduling wm, dw and bp on a Thursday from April as Scenario 2. Thus, the set of similar days *D* is the set of monitored Wednesday days. Then the sets of starting times of *wm*, i.e., $TS_{D}^{\mathrm{wm}}$, that of dw, i.e., $TS_{D}^{\mathrm{dw}}$ and that of bp, i.e., $TS_{D}^{\mathrm{bp}}$, are formed via utilization matrices as follows:$$\begin{array}{c} TS_{D}^{\mathrm{wm}}=\left\{00:00,05:10,05:28,06.00,20:32,23:30\right\}\\ TS_{D}^{\mathrm{dw}}=\left\{16:30,16:40,20:00,22:02,22:30\right\}\\ TS_{D}^{\mathrm{bp}}=\left\{6:00,6:18,6:30,6:52,20:40\right\} \end{array}$$

For each possible starting times of wm, i.e., $t_{s}^{\mathrm{wm}}\in TS_{D}^{\mathrm{wm}}$, that of *dw*, i.e., $t_{s}^{\mathrm{dw}}\in TS_{D}^{\mathrm{dw}}$ and that of bp, i.e., $t_{s}^{\mathrm{bp}}\in TS_{D}^{\mathrm{bp}}$, on similar days in *D*; the number of being chosen as starting time (i.e., $NC_{D}^{\mathrm{wm}}\left(t_{s}^{\mathrm{wm}}\right)$, $NC_{D}^{\mathrm{dw}}\left(t_{s}^{\mathrm{dw}}\right)$) and $NC_{D}^{\mathrm{bp}}\left(t_{s}^{\mathrm{bp}}\right)$), and their probabilities (i.e., $Pr_{D}^{\mathrm{wm}}\left(t_{s}^{\mathrm{wm}}\right)$, $Pr_{D}^{\mathrm{dw}}\left(t_{s}^{\mathrm{dw}}\right)$ and $Pr_{D}^{\mathrm{bp}}\left(t_{s}^{\mathrm{bp}}\right)$ and the corresponding priority levels of the considered day *d* (i.e., ${Pr_{L}}_{d}^{wm}\left(t_{s}^{\mathrm{wm}}\right)$, ${Pr_{L}}_{d}^{dw}\left(t_{s}^{\mathrm{dw}}\right)$, ${Pr_{L}}_{d}^{bp}\left(t_{s}^{\mathrm{bp}}\right)$) are determined as given in Table 9, Table 10 and Table 11.

For each possible combinations of starting times of these appliances, that is, for all possible triplets of ($t_{s}^{\mathrm{wm}},t_{s}^{\mathrm{dw}},t_{s}^{\mathrm{bp}}$) corresponding total electricity cost $C_{d}^{L_{MC}}\left(t_{s}^{wm},t_{s}^{dw},t_{s}^{bp}\right)$, power limit indicator values $I_{d}\left(t_{s}^{wm},t_{s}^{dw},t_{s}^{bp}\right)$ and total priority level values ${Pr_{L}}_{d}^{L_{MC}}\left(t_{s}^{\mathrm{wm}},t_{s}^{\mathrm{dw}},t_{s}^{\mathrm{bp}}\right)$ are represented on the three-axis chart in Figure 10.

In the considered day, through the possible triplets of starting times, triplets with the cheapest cost (16.40 cent), their priority level values and power limit indicator values are given in Table 12 and indicated by blue diamond on the chart in Figure 10. For some of these triplets, the power limit is exceeded. For one of these triplets corresponding daily total power consumption graph is given Figure 11.

On the other hand, through the possible triplets of starting times, the one with the minimum total priority level value (i.e., 1.73) is (05:28,16:30,06:18) (indicated by a red square on the chart in Figure 10). However, the induced cost of this triplet is 23.01 cent, while the maximum power consumption is reached 4427.85 kW.

For this scenario, AS-REMS schedules starting time of wm at (05:28), that of dw at (22:30) and that of bp at (06:18). For the corresponding triplet of starting times, i.e., (05:28,22:30,06:18) (marked as black circle in the chart in Figure 10), the cost is 16.48 cent, the total priority level value is 3.32, while the power limit is not exceeded (maximum power consumption is 4427.85 kW). As it is clear from the Figure 10, this is the closest triplet to the theoretically ideal point. The total power consumption graph of optimal solution is given Figure 11.

Let we consider scheduling dw, wm and bp on a Sunday from September as Scenario 3. Apart from previous scenarios, in this case, user sets specific starting time intervals for wm, dw and bp, such that $T_{s_{interval}}^{wm}=$ [08:44 11:44], $T_{s_{interval}}^{dw}=$ [19:38 23:38], $T_{s_{interval}}^{bp}=$ [20:30 23:30]. The priority level of each of these appliances is set to 0 and the optimal triplet of starting times ($t_{s}^{wm}$, $t_{s}^{dw}$,$t_{s}^{bp}$) must be determined through these intervals, i.e., $t_{s}^{wm}\in T_{s_{interval}}^{wm}$, $t_{s}^{dw}\in T_{s_{interval}}^{dw}$, $t_{s}^{bp}\in T_{s_{interval}}^{bp}$, by considering the cost and power limit parameters.

In the considered day, AS-REMS obtains optimal triplet of starting times as (09:26,21:58,23:00) with the cost of 14.15 cent, while the power limit is not exceeded. The daily power consumption graph is given in Figure 12. Note that, without AS-REMS, wm and dw start to operate at the beginning of their specified starting time interval, such that $t_{s}^{wm}=$ 08:44, $t_{s}^{dw}=$ 19:38 and $t_{s}^{bp}=$ 20:30. In that case, the cost is 23.45 cent while the power limit is exceeded (see Figure 12).

In order to demonstrate AS-REMS’s performance, numerous scenarios (≥500) are designed and the corresponding simulations are carried out. According to the results of these simulations, AS-REMS completely avoids all peak demands exceeding the specified grid power limit by reducing the peak demand levels by approximately 45%. Consequently, smooth and continuous energy from the grid is ensured for the user, while possible maintenance cost of energy provider is reduced. Furthermore, in the simulations, the first preferences of the users are realized by 88% while the electricity costs could be reduced by 39.6%. Thus, users’ appliance usage preferences are sustained at a lower cost while their comfort is kept impressively.

In the simulations of scenarios, sensitivity analysis of computational times are also carried out on a PC with 2.8 GHz CPU, i7 Core, and 16 GB RAM and results are given in Table 13.

The comparison of AS-REMS with the recent studies in the literature from the view of considered parameters and simulation results are given in Table 14 which demonstrates the reasonability and effectiveness of the proposed AS-REMS. Unlike most studies in the literature, in AS-REMS, avoiding peak demand, reducing electricity cost and keeping user comfort are considered simultaneously. Despite this complexity, simulation results of case studies are very satisfactory and also they are much better than the recent works in the literature. This is not surprising, because AS-REMS is based on the users’ appliance usage preferences and habits providing a realistic and high-level user comfort; and real power consumption profiles of appliances are used instead of their average powers assuring to detect even short-term peak demands. In this way, smooth and continuous energy from the grid is also procured.

On the other hand, the hardware configuration of AS-REMS is also constructed to verify the results of simulations and the scenarios are realized on this configuration. At this configuration, a PC with 2.8 GHz CPU, i7 Core, and 16 GB RAM stands for the MC of AS-REMS. Power consumptions of all appliances are measured via Fibaro smart wall plugs connected to the appliances and verified by the Itech IT9121 power meter. PC collects power consumption information of all appliances from smart plugs connected to the appliances via a USB Z-wave stick controller. MC-controllable appliances are equipped with wi-RS232 converter (or Wi-relay module), for starting the operation of appliances. AS-REMS algorithm is implemented via developed C++ software which is also used for simulations of scenarios. Simulation and real application results of scenarios are found to be compatible with each other.

## 4. Conclusions

Due to the increase in the population and the use of technological devices in cities, electricity demand is increasing day by day leading to high depletion of natural resources and pollution of the environment. Besides, peak demand may occur at certain times of the day leading to expensive failures in the grid. This circumstance may also pose serious problems that may affect the public’s social life as disruption of health, education and transportation activities in the cities. Consequently, for both residential users and energy providers sides, avoiding peak demand parameters should also be considered in REM studies.

In this work, a new REM system, namely AS-REMS, is proposed. AS-REMS avoids peak demand and keeps user comfort while reducing electricity costs simultaneously for responding to the expectations of both residential users and energy providers. In AS-REMS, based on the effects of starting times of appliances on user comfort and user attendance during their operations, appliances are divided into two classes such as MC-controllable appliances, whose starting times can be set by MC and user-controllable appliances, whose starting times strictly set by the user even if it is delayable. Use of all appliances are monitored in the considered home for a while for recording users’ appliance usage preferences and habits for each day of the week and for each appliance. Then for each appliance, preferred starting times are determined and prioritized according to the recorded user preferences on similar days. When scheduling, assigned priorities of starting times of MC-controllable appliances are considered for maintaining user comfort, while the tariff rate is considered for reducing the electricity cost. Moreover, expected power consumptions of user-controllable appliances according to user’s usage habits and power consumptions of MC-controllable appliances according to assigned starting times are considered for avoiding peak demand. The practical solution set of the corresponding scheduling problem consists of possible preferred starting time combinations of MC-controllable appliances. Since the numbers of MC-controllable appliances and preferred starting times are limited, the size of practical solution set of the problem is limited. BFCP method whose computational complexity is proportional to the number of candidate solutions, and therefore very suitable for this problem is used to solve it. Besides, the BFCP method is simple to implement and one can add different starting time combinations to the practical solution set, as well as remove some from this set easily.

One future work direction of this work would be to investigate the effects of AS-REMS by integrating it into homes in a neighborhood system. Besides, monitoring the power consumption of household appliances and identifying users’ appliance usage preferences will contribute to future works in research areas, such as improving user comfort and home safety in smart cities.

## Figures and Tables

**Figure 1 sensors-21-03287-f001:**
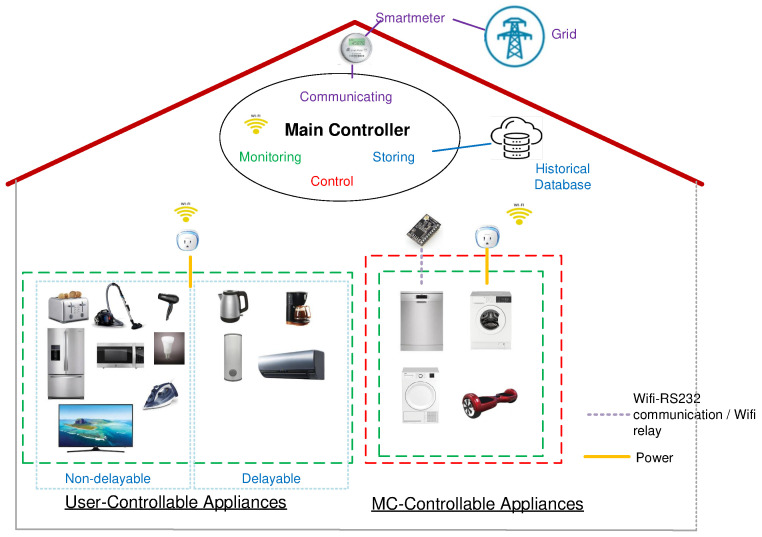
AS-REMS Structure.

**Figure 2 sensors-21-03287-f002:**
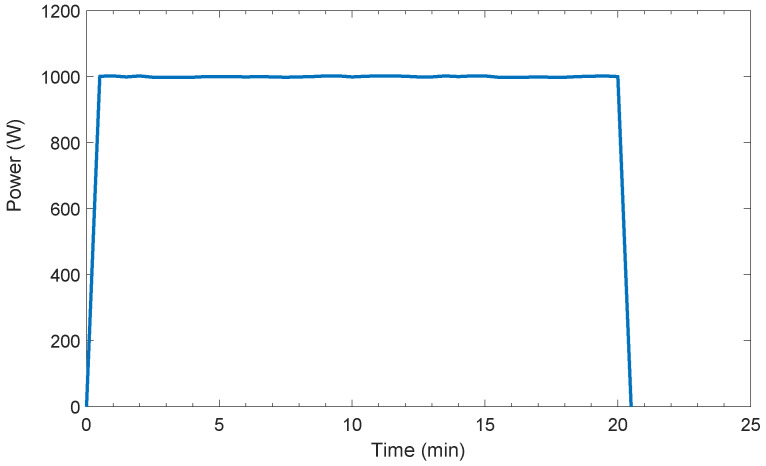
Power consumption profile of a kettle (1000 W and 1.2 lt capacity).

**Figure 3 sensors-21-03287-f003:**
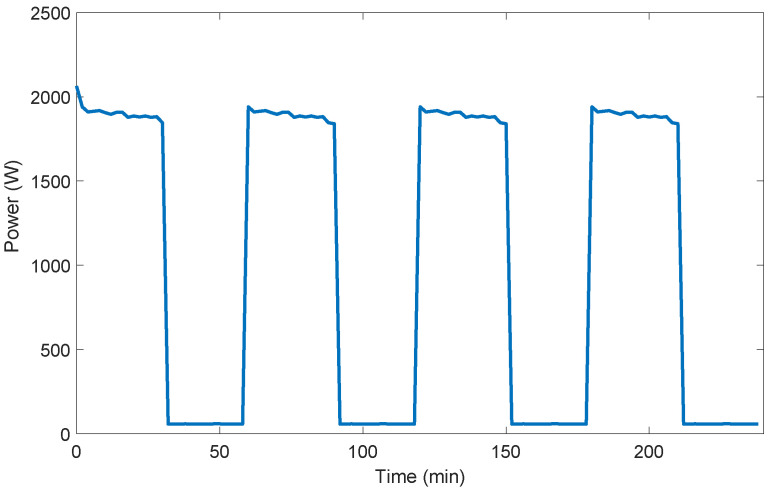
Power consumption profile of an air conditioner (6.74 kW cooling and 7.03 kW heating capacity).

**Figure 4 sensors-21-03287-f004:**
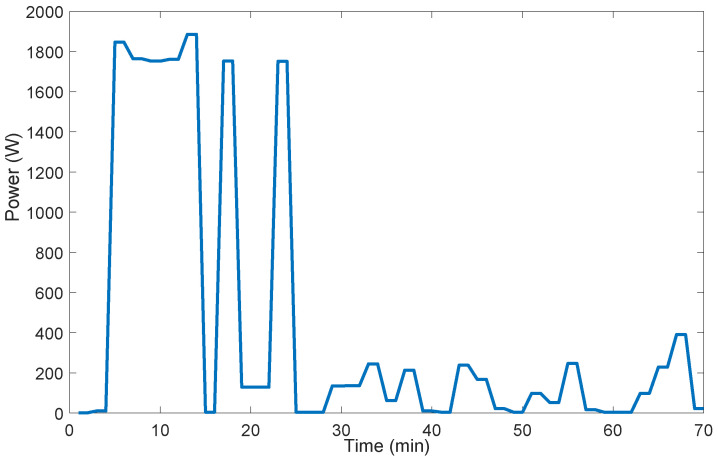
Power consumption profile of a washing machine (7 kg front-load).

**Figure 5 sensors-21-03287-f005:**
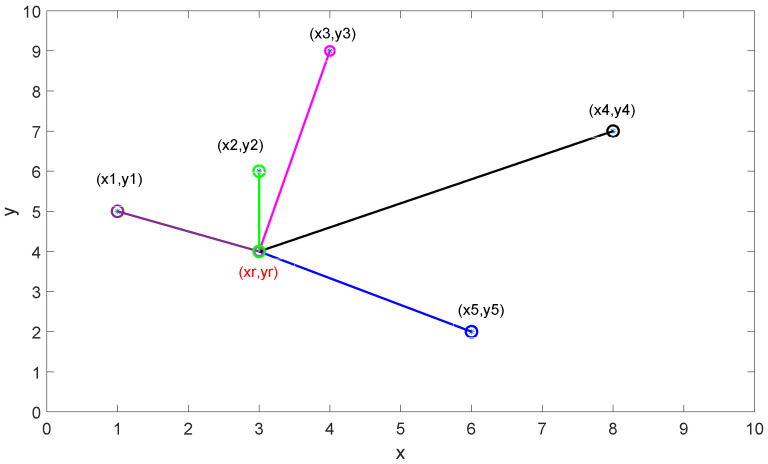
An example application of the BFCP method.

**Figure 6 sensors-21-03287-f006:**
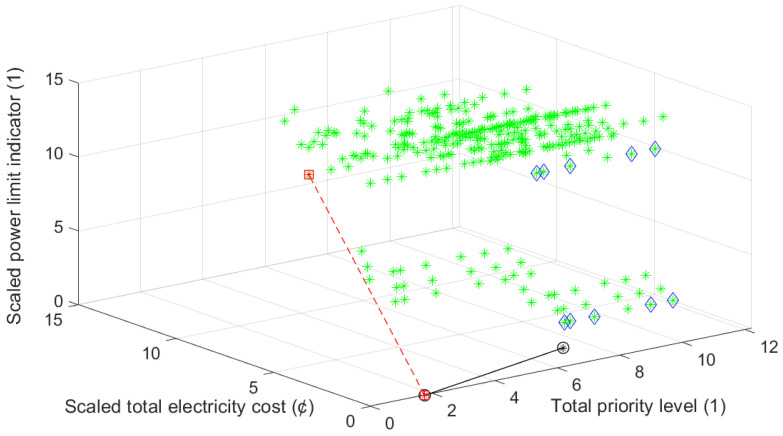
Three-axis chart for possible triplets of ($t_{s}^{\mathrm{wm}},\;t_{s}^{\mathrm{dw}},\;t_{s}^{\mathrm{bp}}$) for Scenario 1.

**Figure 7 sensors-21-03287-f007:**
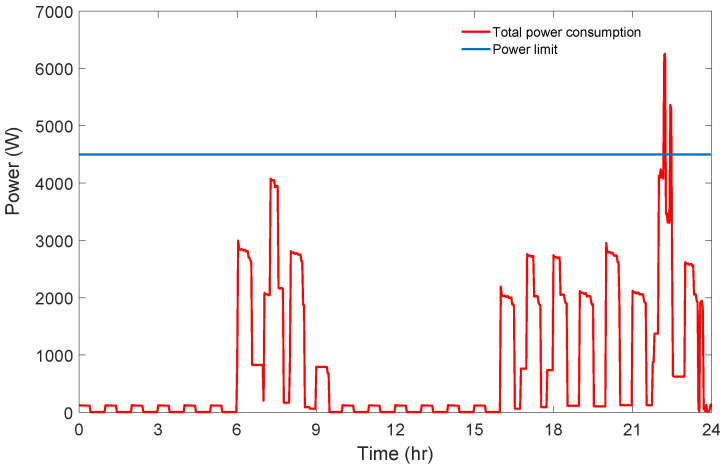
Power consumption graph of the minimum cost solution for Scenario 1.

**Figure 8 sensors-21-03287-f008:**
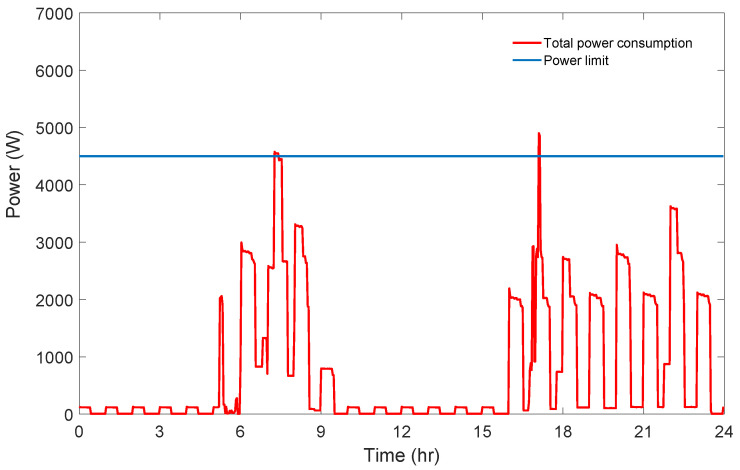
Power consumption graph of the highest priority solution for Scenario 1.

**Figure 9 sensors-21-03287-f009:**
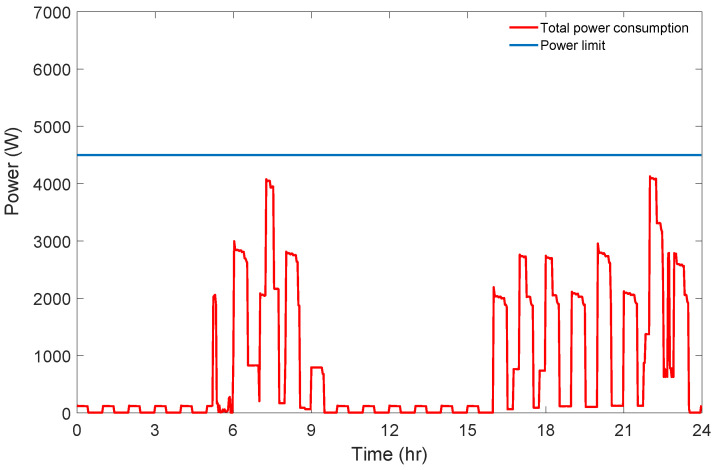
Power consumption graph of the optimal solution for Scenario 1.

**Figure 10 sensors-21-03287-f010:**
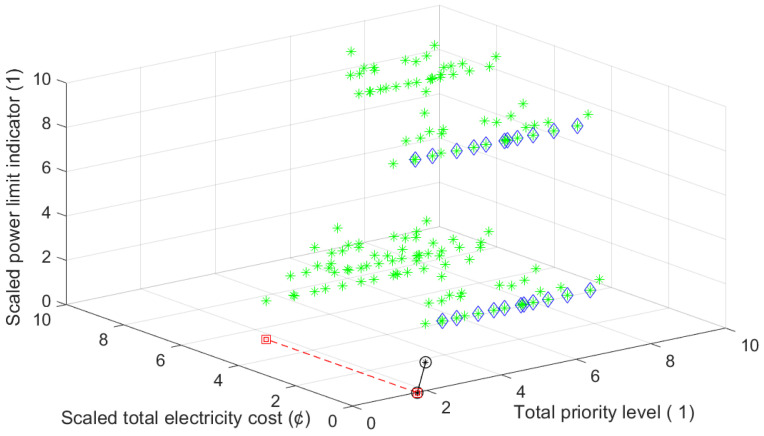
Three-axis chart for possible triplets of ($t_{s}^{\mathrm{wm}},\;t_{s}^{\mathrm{dw}},\;t_{s}^{\mathrm{bp}}$) for Scenario 2.

**Figure 11 sensors-21-03287-f011:**
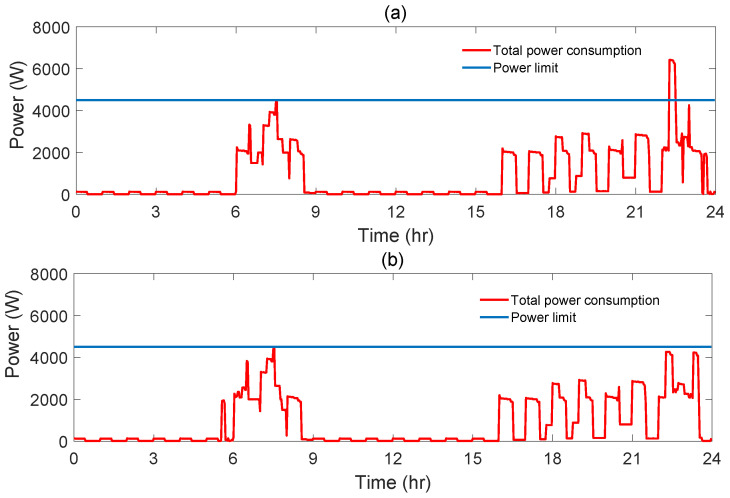
Power consumption graph of minimum cost (**a**) and optimal (**b**) solution for Scenario 2.

**Figure 12 sensors-21-03287-f012:**
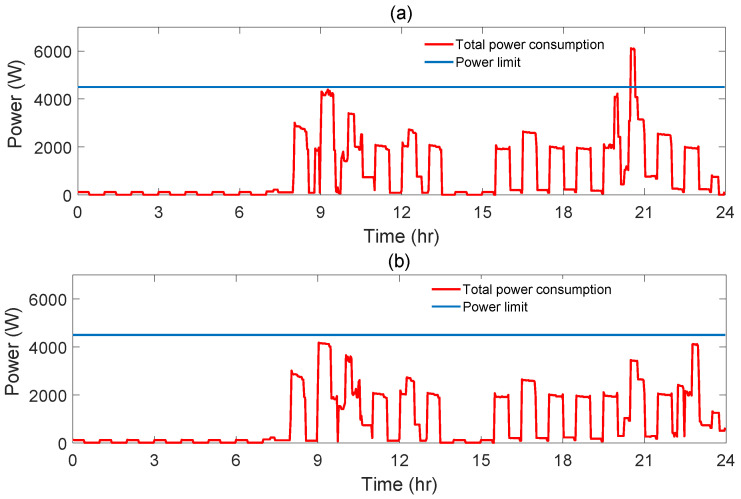
Power consumption graph of both without AS-REMS (**a**) and optimal (**b**) solution for Scenario 3.

**Table 1 sensors-21-03287-t001:** Synthesis of remarkable REM studies in the literature.

Reference	Method	Objective	Description
[5]	Multiobjective optimization programming	Minimizing cost	Schedule some selected appliances using TOU and rated power
[8]	Mixed integer nonlinear programming	Minimizing cost	Schedule the time and energy based appliances
[10]	Binary particle swarm optimization	Minimizing daily electricity bill without effecting comfort	Schedule the shiftable appliances by using TOU and rated power
[12]	Level billing approach with a mathematical model	Minimizing daily bill	Schedule the time shiftable loads
[14]	Intensive quadratic programming	Minimizing cost and peak	Flatten the power consumption by using PV power
[20]	Intensive based energy optimization	Minimizing the electricity bill	Schedule the shiftable appliances by using rated power
[24]	Rolling wave planning	Minimizing cost and peak	Control controllable appliances by using the real power consumption

**Table 2 sensors-21-03287-t002:** Types of classifications at REM studies in the literature.

Appliance	NO,FTB	C,CBE,EBE	UnC	Sh,Sc	USh,USc	AS-REMS
WM		3,23,24		2,7,8,11,		MCC
				13,14,15,19,20		
DW		23,24		2,8,12,13,		MCC
				14,15,20,21		
Thumble dryer		16		7,13,14,20,21		MCC
Battery Powered						MCC
Pool pump		2,8,11,15,22				MCC
Coffee Machine					7	dUSC
Kettle					2	dUSC
Water Heater			7,11,12		7	dUSC
AC	8,13	15,22			2	dUSC
Lights	8,13,15,20		22,23,24		2,7,11	ndUSC
TV	15		23,24		2,7,11,19	ndUSC
Blender					2,11	ndUSC
Hair dryer					11	ndUSC
Microwave oven	3				7	ndUSC
Vacuum cleaner	8,20				2,7,11	ndUSC
Refrigerator	8	22	12		11,24	ndUSC

**Table 3 sensors-21-03287-t003:** Pricing tariff rates.

Duration	Cost (Euro/kWh)	$P_{\lim}$
6:00 a.m.–5:00 p.m.	0.094	4500 W
5:00 p.m.–10:00 p.m.	0.136	4500 W
10:00 p.m.–6:00 a.m.	0.059	4500 W

**Table 4 sensors-21-03287-t004:** Priority level values for $t_{s}^{\mathrm{wm}}$ for Scenario 1.

$t_{s}^{\mathrm{wm}}$	${\mathrm{NC}}_{D}^{\mathrm{wm}}\left(t_{s}^{\mathrm{wm}}\right)$	${\mathrm{Pr}}_{D}^{\mathrm{wm}}\left(t_{s}^{\mathrm{wm}}\right)$	${{\mathrm{Pr}}_{L}}_{d}^{\mathrm{wm}}\left(t_{s}^{\mathrm{wm}}\right)$
05:12	18	34.62	1
00:00	9	17.31	2
20:28	9	17.31	3
23:30	6	11.54	4
06:00	4	7.69	5
05:28	2	3.85	6
23:38	2	3.85	7
05:00	2	3.85	8

**Table 5 sensors-21-03287-t005:** Priority level values for $t_{s}^{\mathrm{dw}}$ for Scenario 1.

$t_{s}^{\mathrm{dw}}$	${\mathrm{NC}}_{D}^{\mathrm{dw}}\left(t_{s}^{\mathrm{dw}}\right)$	${\mathrm{Pr}}_{D}^{\mathrm{dw}}\left(t_{s}^{\mathrm{dw}}\right)$	${{\mathrm{Pr}}_{L}}_{d}^{\mathrm{dw}}\left(t_{s}^{\mathrm{dw}}\right)$
16:40	16	30.77	1
22:00	9	17.31	2
17:04	8	15.38	3
22:32	8	15.38	4
16:30	5	9.62	5
19:12	4	7.69	6
20:00	2	3.85	7

**Table 6 sensors-21-03287-t006:** Priority level values for $t_{s}^{\mathrm{bp}}$ for Scenario 1.

$t_{s}^{\mathrm{bp}}$	${\mathrm{NC}}_{D}^{\mathrm{bp}}\left(t_{s}^{\mathrm{bp}}\right)$	${\mathrm{Pr}}_{D}^{\mathrm{bp}}\left(t_{s}^{\mathrm{bp}}\right)$	${{\mathrm{Pr}}_{L}}_{d}^{\mathrm{bp}}\left(t_{s}^{\mathrm{bp}}\right)$
06:50	17	32.69	1
06:00	13	25.00	2
20:38	8	15.38	3
06:30	6	11.54	4
06:20	6	11.54	5
21:52	2	3.85	6

**Table 7 sensors-21-03287-t007:** For the possible triplets of starting times; priority level, electricity cost, power limit indicator, scaled priority level, scaled power limit indicator values and the corresponding values of objective function of BFCP Algorithm (Algorithm 2) for Scenario 1.

($t_{s}^{wm},t_{s}^{dw},t_{s}^{bp}$)	${{Pr}_{L}}_{d}^{L_{MC}}\left(.\right)$	$C_{d}^{L_{MC}}$(.)	$I_{d}\left(.\right)$	$C_{dscaled}^{L_{MC}}\left(.\right)$	$I_{dscaled}$	$d_{i}\left(.\right)$
(05:12,16:40,06:50)	1.73	13.44	1	5.94	12.21	13.69
(00:00,16:40,06:50)	2.45	13.44	1	5.94	12.21	13.79
(20:28,16:40,06:50)	3.32	15.83	1	8.46	12.21	15.22
(23:30,16:40,06:50)	4.24	13.44	1	5.94	12.21	14.22
(06:00,16:40,06:50)	5.20	14.50	1	7.06	12.21	15.03
(-,-,-)	-	-	-	-	-	-
(05:12,22:32,21:52)	7.28	9.45	0	1.73	1.73	7.68
(-,-,-)	-	-	-	-	-	-
(05:28,20:00,21:52)	11.00	12:20	1	4.63	12.21	17.07
(23:38,20:00,21:52)	11.58	12.11	1	4.54	12.21	17.42
(05:00,20:00,21:52)	12.21	12.11	1	4.54	12.21	17.85

**Table 8 sensors-21-03287-t008:** For the triplets of starting times with the minimum cost; priority level, electricity cost, power limit indicator, scaled priority level, scaled power limit indicator values and the corresponding values of objective function of BFCP Algorithm 2 for Scenario 1.

($t_{s}^{wm},t_{s}^{dw},t_{s}^{bp}$)	${{Pr}_{L}}_{d}^{L_{MC}}\left(.\right)$	$C_{d}^{L_{MC}}$(.)	$I_{d}\left(.\right)$	$C_{dscaled}^{L_{MC}}\left(.\right)$	$I_{dscaled}$	$d_{i}\left(.\right)$
(05:12,22:00,21:52)	6.40	9.45	1	1.73	12.21	13.89
(00:00,22:00,21:52)	6.63	9.45	1	1.73	12.21	14.00
(23:30,22:00,21:52)	7.48	9.45	1	1.73	12.21	14.42
(23:38,22:00,21:52)	9.43	9.45	1	1.73	12.21	15.51
(05:00,22:00,21:52)	10.20	9.45	1	1.73	12.21	16.00
(05:12,22:32,21:52)	7.28	9.45	0	1.73	1.73	7.68
(-,-,-)	-	-	-	-	-	-
(23:38,22:32,21:52)	10.05	9.45	0	1.73	1.73	10.34
(05:00,22:32,21:52)	10.77	9.45	0	1.73	1.73	11.05

**Table 9 sensors-21-03287-t009:** Priority level values for $t_{s}^{\mathrm{wm}}$ for Scenario 2.

$t_{s}^{\mathrm{wm}}$	${\mathrm{NC}}_{D}^{\mathrm{wm}}\left(t_{s}^{\mathrm{wm}}\right)$	${\mathrm{Pr}}_{D}^{\mathrm{wm}}\left(t_{s}^{\mathrm{wm}}\right)$	${{\mathrm{Pr}}_{L}}_{d}^{\mathrm{wm}}\left(t_{s}^{\mathrm{wm}}\right)$
05:28	18	34.62	1
23.30	11	21.15	2
20.32	9	17.31	3
00:00	6	11.54	4
05:10	6	11.54	5
06:00	2	3.85	6

**Table 10 sensors-21-03287-t010:** Priority level values for $t_{s}^{\mathrm{dw}}$ for Scenario 2.

$t_{s}^{\mathrm{dw}}$	${\mathrm{NC}}_{D}^{\mathrm{dw}}\left(t_{s}^{\mathrm{dw}}\right)$	${\mathrm{Pr}}_{D}^{\mathrm{dw}}\left(t_{s}^{\mathrm{dw}}\right)$	${{\mathrm{Pr}}_{L}}_{d}^{\mathrm{dw}}\left(t_{s}^{\mathrm{dw}}\right)$
16:30	15	28.85	1
22.02	15	28.85	2
22.30	14	26.92	3
16.40	6	11.54	4
20.00	2	3.85	5

**Table 11 sensors-21-03287-t011:** Priority level values for $t_{s}^{\mathrm{bp}}$ for Scenario 2.

$t_{s}^{\mathrm{bp}}$	${\mathrm{NC}}_{D}^{\mathrm{bp}}\left(t_{s}^{\mathrm{bp}}\right)$	${\mathrm{Pr}}_{D}^{\mathrm{bp}}\left(t_{s}^{\mathrm{bp}}\right)$	${{\mathrm{Pr}}_{L}}_{d}^{\mathrm{bp}}\left(t_{s}^{\mathrm{bp}}\right)$
06:18	20	38.46	1
06:00	11	21.15	2
06:30	9	17.31	3
20:40	6	11.54	4
06:52	6	11.54	5

**Table 12 sensors-21-03287-t012:** For the triplets of starting times with the minimum cost, priority level, electricity cost, power limit indicator values for Scenario 2.

($t_{s}^{\mathrm{wm}},t_{s}^{\mathrm{dw}}$)	${{\mathrm{Pr}}_{L}}_{d}^{L_{\mathrm{MC}}}\left(.\right)$	$C_{d}^{L_{\mathrm{MC}}}\left(.\right)$	$I_{d}\left(.\right)$
(23:30,22:02,06:18)	3.00	16.40	1
(00:00,22:02,06:18)	4.58	16.40	1
(05:10,22:02,06:18)	5.48	16.40	1
(23:30,22:30,06:18)	3.74	16.40	1
(00:00,22:30,06:18)	5.10	16.40	1
(05:10,22:30,06:18)	5.92	16.40	1
(23:30,22:02,06:00)	3.46	16.40	1
(00:00,22:02,06:00)	4.90	16.40	1
(-,-,-)	-	-	-
(05:10,22:30,06:52)	7.68	16.40	1

**Table 13 sensors-21-03287-t013:** Sensitivity of computational times with respect to the number of possible starting time combinations.

# of Combinations of Starting Times ($|{\mathrm{TS}}_{D}^{L_{\mathrm{MC}}}|$)	Computational Time
{12}	1.57 s
{64}	3.33 s
{128}	5.45 s
{256}	9.14 s
{384}	11.54 s
{512}	17.39 s

**Table 14 sensors-21-03287-t014:** Comparision of AS-REMS with the recent REM studies in the literature.

Method	Cost Minimization	Peak Reduction	User Comfort
Incentive-based energy optimization method [20]	6.2%	21%	no value
Intensive quadratic programming approach [14]	10%	44%	no value
Level billing approach [12]	13–25%	not considered	only financial satisfaction
Appliance based Rolling Wave Planning algorithm [24]	13–24%	38–53%	no value
Binary particle swarm optimization [10]	32.8%	-	66%
AS-REMS	39.6%	45%	88%

## Data Availability

The data that support the findings of this study are available from the corresponding author, Hanife Apaydin-Özkan, upon reasonable request.

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
