# Peer review of "An Appliance Scheduling System for Residential Energy Management"

_sensors, 2021, doi:10.3390/s21093287_

Round 1
Reviewer 1 Report
Dear Authors
Your idea of reducing electrical energy consumption based on a programmed use control is very good, especially in this time of COVID, when people pass or must stay at home. AS-REMS aims to schedule MC-controllable appliances by keeping user comfort and reducing electricity cost as well as avoiding peak demand.
In the first sections, the text is piled up, it should be organized in a better way. Improve the quality of figures, label, size, units, etc.
You analyze a case study, propose an algorithm, and via simulations obtain the results, you must explain the algorithm in more detail.
Likewise, it would be healthy for the algorithm to be validated in some other way, for example with specialized tools for this type of analysis, even the results that would be obtained would be better than the current ones.
Author Response
Response to Reviewer 1 Comments
Your idea of reducing electrical energy consumption based on a programmed use control is very good, especially in this time of COVID, when people pass or must stay at home. AS-REMS aims to schedule MC-controllable appliances by keeping user comfort and reducing electricity cost as well as avoiding peak demand.
Thank you for your comments. We have gone through your comments carefully and tried our best to address them one by one. We believe that the manuscript has been improved accordingly.
Point 1. In the first sections, the text is piled up, it should be organized in a better way. Improve the quality of figures, label, size, units, etc.
Response 1. In the first section, we organized the literature survey again and added a table (Table 1 at page 3) consisting of the synthesis of remarkable REM studies in the literature.
Point 2. Improve the quality of figures, label, size, units, etc.
Response 2. We checked all figures and then corrected all labels, size and units.
Point 3. You analyze a case study, propose an algorithm, and via simulations obtain the results, you must explain the algorithm in more detail.
Response 3. We thank the reviewer for pointing out this lack of clarity. In order to clarify this concept, we revised the Section 2.4 ( Scheduling Procedure) in detail and added the AS-REMS algorithm scheduling scheme in details.
Point 4. Likewise, it would be healthy for the algorithm to be validated in some other way, for example with specialized tools for this type of analysis, even the results that would be obtained would be better than the current ones.
Response 4. In order to validate AS-REMS in another way, the installed actual hardware configuration is also introduced in the revised article. Note that, simulation and real application results of AS-REMS are found to be compatible with each other.
On the other hand, we would like to underline the Table 14 at page 20, which represents the comparision of simulation results of AS-REMS and similar REM studies in the literature.

Reviewer 2 Report
In my opinion, the paper is in general interesting and nice to read. The manuscript deserves to be published only once the authors fix the following issues.
Literature review
- The main contributions of the paper are clearly described. Nevertheless, from the current manuscript it is not grasp understanding the novelty of the work. The authors should better highlight the innovative aspects of their work in the manuscript.
Problem formulation
- The load dynamics in Section 2 shows the modeling of control based loads. Several recent scientific studies on energy scheduling (e.g., https://doi.org/10.1109/TASE.2020.2986269, https://doi.org/10.3390/en7095787, documents that could be cited in the text), show that loads could be classified into non-controllable, controllable comfort-based loads, and controllable energy based loads. Do the authors handle all these loads? The Authors should comment this point.
- The authors should clearly characterize the overall problem that they intend to solve. What type of decision variables (i.e. integer, real, etc) and how many? How many constraints (bounding, inequality, equality)?
- The authors should clarify how they handle the uncertainty of parameters. Several recent scientific studies on power grid (e.g., https://doi.org/10.1109/TSG.2018.2863049, documents that could be referenced in the text), show that robust optimization instead of classical optimization is a viable technique to deal with uncertainty of parameters. The Authors should comment this point.
Algorithm resolution
- A preliminary section about the description of BFCP method could help the readers to better understand the proposed algorithm.
Case study
- There is no sensitivity analysis in the paper. Is it reasonable?
Minor
- The authors should check that all the used acronyms are explained.
- Mainly the English is good and there are only a few typos. However the paper should be carefully rechecked.
Author Response
Response to Reviewer 2 Comments
In my opinion, the paper is in general interesting and nice to read. The manuscript deserves to be published only once the authors fix the following issues.
Thank you for your valuable comments. We have read your comments carefully and tried our best to address them one by one, especially in terms of providing a more rigorous model evaluation.
Literature review
Point 1. The main contributions of the paper are clearly described. Nevertheless, from the current manuscript it is not grasp understanding the novelty of the work. The authors should better highlight the innovative aspects of their work in the manuscript.
Response 1. In the first section of the revised manuscript, the literature survey is re-organized. A table (Table 1 at page 3) consisting of the synthesis of remarkable REM studies in the literature is given. The main contributions of AS-REMS is also addressed and itemized (lines from 120 to 129) in the revised manuscript.
Point 2. The load dynamics in Section 2 shows the modeling of control based loads. Several recent scientific studies on energy scheduling (e.g., https://doi.org/10.1109/TASE.2020.2986269, https://doi.org/10.3390/en7095787, documents that could be cited in the text), show that loads could be classified into non-controllable, controllable comfort-based loads, and controllable energy based loads. Do the authors handle all these loads? The Authors should comment this point.
Response 2. The papers you suggested are added to the revised manuscript.
https://doi.org/10.1109/TASE.2020.2986269
- Hosseini, S.M.; Carli, R.; Dotoli, M. Robust Optimal Energy Management of a Residential Microgrid Under Uncertainties on Demand and Renewable Power Generation. IEEE Transactions on Automation Science and Engineering 2021, 18, 618–637. doi:10.1109/TASE.2020.2986269.
https://doi.org/10.3390/en7095787
- Barbato, A.; Capone, A. Optimization Models and Methods for Demand-Side Management of Residential Users: A Survey. Energies 2014, 7, 5787–5824. doi:10.3390/en7095787.
In the first study, appliances were classified as controllable load (CL) and noncontrollabble load (NCL). This controllable load was subsequently divided into two subclasses; energy based controllable and confort based controllable. Some appliances were given as an example of these loads, for example; light (non-controllable load), pump (energy based controllable), air condonditioner and refrigerator (comfort based controllable ).
In the second study, appliances were classified as fixed appliances, shiftable appliances and elastic appliances. This elastic appliance was subsequently divided into two subclasses; energy based elastic and confort based elastic. Some appliances were given as an example of these appliancess, for example; light, TV (fixed appliances), pool pump (energy based elastic), air condonditioner and refrigerator (comfort based elastic ), washing machine and dishwasher ( shiftable appliances).
If we compare our classification methot with these two studies, it can be seen that the similar approach is used.
In these studies, the operation of lights, TV, airconditioner, refrigerator depends on user preferences and their operations effect the user comfort directly. Washing machine, dishwasher and pumps were classified as shiftable or controllable. In our study, we handle all these appliances. Lights, TV, airconditioner and refrigerator were classified as user controllable. Main controller can not control the operation of these appliances. Washing machine, dishwasher and pumps were classified as MC- controllable appliances.
Main controller schedules MC-controllable appliances at the beginning of each day under the aims of avoiding peak demand and reducing electricity cost while keeping user comfort.
Point 3. The authors should clearly characterize the overall problem that they intend to solve. What type of decision variables (i.e. integer, real, etc) and how many? How many constraints (bounding, inequality, equality)?
Response 3. We thank the reviewer for pointing out this lack of clarity. In order to clarify this concept, we revised the section 2.3 (Scheduling Parameters) and 2.4 (Scheduling Procedure) and characterize the overall problem by explaining the parameters (lines from 290 to 308) , objective function (lines from 308 to 310) and the constraints (lines from 326 to 328).
Point 4. The authors should clarify how they handle the uncertainty of parameters. Several recent scientific studies on power grid (e.g., https://doi.org/10.1109/TSG.2018.2863049, documents that could be referenced in the text), show that robust optimization instead of classical optimization is a viable technique to deal with uncertainty of parameters. The Authors should comment this point.
Response 4. The paper you suggested is cited in the revised manuscript.
The methods of eliminating uncertainities are also explained in Section 2.3 (Scheduling Parameters) (lines from 243 to 250 and lines from 273 to 277)
Algorithm resolution
Point 5. A preliminary section about the description of BFCP method could help the readers to better understand the proposed algorithm.
Response 5. Thank you very much for pointing out this lack of clarity. A new section ( Section 2.3. Brute Force Closest Pair Method) is added to the revised manuscript.
Case study
Point 6. There is no sensitivity analysis in the paper. Is it reasonable?
Response 6. In the simulations of AS-REMS, time slot duration is taken as 2 min which may catch even short term power variations. Sensitivity analysis of corresponding computational times is given in Table 13.
Minor
Point 7. The authors should check that all the used acronyms are explained.
Response 7. Thank you, it is done.
Point 8. Mainly the English is good and there are only a few typos. However the paper should be carefully rechecked.
Response 8. Thank you, we carefully revised manuscript with the help of a native speaker.

Reviewer 3 Report
The topic is very interesting. The problem is much debated in the literature in the last years. In the paper, the authors proposed an appliance scheduling based residential energy management system which avoids peak demand and keeps user comfort while reducing electricity cost. The obtained results are encouraging. However, there are some issues that the author should consider to improve the structure and quality of the paper:
- In Abstract the authors assert "Simulation results of case studies show reasonability and effectiveness of AS-REMS.” It would be useful for readers to be indicated the values which to strengthen these statements.
- In the first part of the paper, the authors present various approaches from the literature. Maybe, a synthesis of the solutions proposed in the literature depending on the type of analysis, which to highlight more clearly the advantages and disadvantages, is useful for readers. This synthesis can be given as a table.
- The title of the paper is "An Appliance Scheduling System for Residential Energy Management". A general AS-REMS structure was presented in Fig. 1. The authors should clearly specify whether the hardware part has been implemented in the system because only the software part is presented in the paper.
- The authors should presents the typical profiles of all appliances considered in the case studies. It is not clear if these are similar with those from the first part of the paper.
Author Response
Response to Reviewer 3 Comments
The topic is very interesting. The problem is much debated in the literature in the last years. In the paper, the authors proposed an appliance scheduling based residential energy management system which avoids peak demand and keeps user comfort while reducing electricity cost. The obtained results are encouraging. However, there are some issues that the author should consider to improve the structure and quality of the paper:
Thank you for your valuable comments.We have read your comments carefully and addressed each of them particularly, especially in terms of providing a more rigorous model evaluation.
Point 1. In Abstract the authors assert "Simulation results of case studies show reasonability and effectiveness of AS-REMS.” It would be useful for readers to be indicated the values which to strengthen these statements.
Response 1. Thank you very much for the comment. In the revised manuscript numerical results are added in abstract (lines from 15 to 18) to show the reasonability and effectiveness of AS-REMS.
Point 2. In the first part of the paper, the authors present various approaches from the literature. Maybe, a synthesis of the solutions proposed in the literature depending on the type of analysis, which to highlight more clearly the advantages and disadvantages, is useful for readers. This synthesis can be given as a table.
Response 2. In the first section of the revised manuscript, the literature survey is re-organized. A table (Table 1) consisting of the synthesis of remarkable REM studies in the literature. The main contributions of AS-REMS is also addressed and itemized in the revised manuscript.
On the other hand, we would like to take attention to Table 13 which represents the comparision of simulation results of AS-REMS and REM studies in the literature.
Point 3. The title of the paper is "An Appliance Scheduling System for Residential Energy Management". A general AS-REMS structure was presented in Fig. 1. The authors should clearly specify whether the hardware part has been implemented in the system because only the software part is presented in the paper.
Response 3. Thank you very much for pointing out this important point. The hardware configuration of AS-REMS is also constructed. In the revised manuscript, this configuration and the specifications of this configuration are also introduced (lines from 450 to 460).
Point 4. The authors should presents the typical profiles of all appliances considered in the case studies. It is not clear if these are similar with those from the first part of the paper.
Response 4. Thank you for this justified warning. The appliances whose power profiles are illustrated in Section 2 are same as the ones used in the simulations given in the Section 3. In order to clarify this, we gave the specifications of considered appliances in the caption of their power consumption graphs (Fig 2, Fig 3 and Fig 4) in Section 2 and in Section 3 (lines from 333-342).

Round 2
Reviewer 1 Report
Dear authors
The article is much better.
Congratulations ...
Reviewer 2 Report
Previous comments and concerns have been sufficiently addressed. In the revised paper several improvements have been added.